# A Narrative Review of Health Status and Healthcare Delivery in the Oil and Gas Industry: Impacts on Employees, Employers, and Local Communities

**DOI:** 10.3390/healthcare11212888

**Published:** 2023-11-02

**Authors:** Jordan L. Fox, Tiana Gurney, Srinivas Kondalsamy-Chennakesavan, Thomas A. H. Pols, Haitham Tuffaha, Anton Pak, Matthew McGrail

**Affiliations:** 1Rural Clinical School, The University of Queensland, Rockhampton 4700, Australia; m.mcgrail@uq.edu.au; 2Rural Clinical School, The University of Queensland, Toowoomba 4350, Australia; t.gurney@uq.edu.au (T.G.); s.kondalsamychennakes@uq.edu.au (S.K.-C.); 3Shell Australia Pty Ltd., Brisbane 4000, Australia; thomas.pols@shell.com; 4Centre for the Business and Economics of Health, The University of Queensland, Brisbane 4072, Australia; h.tuffaha@uq.edu.au (H.T.); a.pak@uq.edu.au (A.P.)

**Keywords:** remote healthcare, telehealth, telemedicine, resource sector, well-being, primary care, emergency response, trauma, fly-in fly-out, public–private partnerships

## Abstract

Workers in the oil and gas industry are exposed to numerous health risks, ranging from poor health behaviours to the possibility of life-threatening injuries. Determining the most appropriate models of healthcare for the oil and gas industry is difficult, as strategies must be acceptable to multiple stakeholders, including employees, employers, and local communities. The purpose of this review was to broadly explore the health status and needs of workers in the oil and gas industry and healthcare delivery models relating to primary care and emergency responses. Database searches of PubMed, EMBASE, CINAHL, PsycINFO, and Scopus were conducted, as well as grey literature searches of Google, Google Scholar, and the International Association of Oil and Gas Producers website. Resource-sector workers, particularly those in ‘fly-in fly-out’ roles, are susceptible to poor health behaviours and a higher prevalence of mental health concerns than the general population. Evidence is generally supportive of organisation-led behaviour change and mental health-related interventions. Deficiencies in primary care received while on-site may lead workers to inappropriately use local health services. For the provision of emergency medical care, telehealth and telemedicine lead to favourable outcomes by improving patient health status and satisfaction and reducing the frequency of medical evacuations.

## 1. Introduction

The oil and gas industry may expose its employees to a range of risks to their health. These include: (1) occupational risks and hazards, including toxins [1]; (2) environmental risks such as heat/cold exposure or solitary work; and (3) less-healthy work patterns such as shift work and ‘fly-in fly-out’ (FIFO) work arrangements. Given the various potential health risks associated with working in the industry, it is important that oil and gas workers have access to a diverse mix of healthcare services, including healthcare for emergency or trauma, acute illnesses, occupational health and safety, primary care, and mental health and well-being.

While health needs play a substantial role in dictating the healthcare requirements of oil and gas workers, the geographical location and environment in which they work also plays a significant role in determining access to appropriate healthcare. Previous research has focussed strongly on healthcare for offshore oil and gas workers [2,3,4], due to their geographical isolation and the logistics associated with patient retrieval following illness or injury. Although caring for the offshore workforce is of critical importance, onshore workers face many of the same circumstances as offshore workers, often working in small and remote communities far from larger service towns, as well as being on FIFO or other rotational work arrangements. Therefore, by comprehensively exploring healthcare for oil and gas workers, information can be generated regarding ideal models of healthcare for the industry, as well as strategies which can be applied to onshore and offshore contexts.

Healthcare models must be effective, efficient, and acceptable to both employees and employers; however, healthcare for oil and gas workers may also impact wider groups of stakeholders, such as health service employees and residents within communities hosting oil and gas workers. Oil and gas workers may need to access locally available healthcare services while at work, via either primary care or emergency services. In doing so, they may put additional strain on already-pressured health services or affect the quality and availability of healthcare for local residents, particularly in more remote areas. Although a number of reviews have been conducted relating to health and healthcare for oil and gas workers, these reviews have each focused on specific aspects of health/healthcare such as health behaviours [3], offshore medical evacuations [4], and employee mental health [5]. While these topics are all valuable in helping to promote the favourable health status and safety of workers, it is the interplay between health needs, healthcare availability, and community needs, among other factors, which ultimately dictates how oil and gas organizations should deliver healthcare and manage the health of their workers. Therefore, it is necessary to comprehensively explore health needs and healthcare delivery for oil and gas workers in onshore and offshore contexts along with any potential flow-on effects to communities nearby where they work. As such, the aim of this review was to broadly explore the health status and needs of workers in the oil and gas industry, and healthcare delivery models including, but not limited to, primary care and emergency responses, and innovative models of care. A secondary aim was to understand how various healthcare models may impact the communities in which oil and gas workers are based or come to work. Although having appropriate occupational health strategies is a key deliverable for the oil and gas industry, it was not a focus of the review, given that it is largely driven by legislative requirements.

## 2. Materials and Methods

A narrative review methodology was employed due to the broad scope of information sought and topics of interest. Where insufficient evidence was available from the oil and gas industry, evidence was drawn from similar contexts, such as the wider resource sector. This approach was taken in order to be able to consider healthcare models which have demonstrated effectiveness within the oil and gas industry, as well as to explore models from similar industries which could be considered. As such, it was not possible to identify a single targeted research question or apply explicit inclusion and exclusion criteria, which would be required for other review types such as a systematic or scoping review [6,7]. Nevertheless, a comprehensive search strategy was developed to ensure information relevant to the review was effectively detected.

### Search Strategy

Due to the breadth of the topic and the evidence of interest, multiple targeted searches were conducted to identify the relevant literature. The topics of each search, databases searched, and keywords used to construct the searches are provided in Table 1, and the complete search strategies used for each database are provided as supplementary online material (Appendix A). Database searches were conducted using a range of medical and health-focussed databases, including PubMed, EMBASE, CINAHL, PsycINFO, and Scopus. Additionally, a grey literature search was conducted relating to healthcare in the oil and gas industry. Google and Google Scholar (grey literature) were the sole databases searched for evidence on public–private partnerships, as the majority of the evidence and available data are published outside of academic journals. Lastly, reports from the International Association of Oil and Gas Producers (IOGP) were searched via their website (https://www.iogp.org/) to help identify established guidelines and recommendations for the provision of healthcare within the oil and gas industry. Reference lists of included articles were also searched.

All searches were restricted to articles published in English within the last ten years; however, older articles identified via searching reference lists were included if they were pertinent to the review. Both primary and secondary sources were included in the review in order to identify areas of interest in which a substantial body of evidence was already available (e.g., systematic reviews).

The lead author was responsible for searching and screening articles. All articles retrieved from electronic databases were screened, along with the first 50 pages of results (if sufficient results were retrieved, sorted by relevance) for all grey literature searches. The lead author read all titles and abstracts before relevant full-text items were reviewed and the key findings from all relevant articles summarised. Based on the key findings from the included studies, the lead author, with input from the wider authorship team, developed a narrative summary of the main themes and findings, which are presented in the subsequent sections of this review.

## 3. Results and Discussion

### 3.1. Search Results

A total of 14,785 articles were retrieved via database searching and an additional two IOGP reports were retrieved. After screening all articles and grey literature, 118 documents were determined relevant to the review. Key findings from each relevant document are provided in Appendix A, and a narrative summary of the themes and important findings arising from the review is provided below. Due to the narrative review methodology utilized, Appendix A is not intended to be a detailed and exhaustive table of results from individual studies, but rather a resource accompanying the review, to allow readers to further explore the topic.

### 3.2. Topic 1: Health Status and Needs

The health status of oil and gas workers is important to ascertain, as this insight will drive how healthcare is delivered. For the purposes of this review, data are drawn from the entire resource sector, inclusive of the broader mining sector. While mine work and contexts largely differ from the oil and gas industry, there are a number of similarities in the populations and workforce design which make these data somewhat comparable. Specifically, both industries are typically male-dominated, workers may be exposed to monotonous/repetitive tasks, and both have a heavy reliance on a FIFO workforce. All of these factors contribute to differing health behaviours, health status, and occupational risks. Mine workers often report low exercise levels, poor nutrition, higher levels of stress, and increased fatigue, which all correlate with low job satisfaction [8]. In addition, health status appears more adversely affected where employees work in remote environments. Amongst offshore oil and gas workers, more than 70% were reported to be overweight or obese, 67% had poor sleep quality, over 50% engaged in risky levels of alcohol consumption, and 15% had a long-term illness [3].

#### 3.2.1. FIFO Work Arrangements

One of the most widely researched health and safety concerns within the resource sector is the array of consequences relating to FIFO work arrangements. A recent systematic review [2] found that FIFO arrangements in the oil and gas industry contribute to poor physical health as well as higher levels of psychological distress, depression, and anxiety.

#### 3.2.2. Impacts of FIFO Work Arrangements on Health Behaviours

In terms of physical health, it has been reported that while working away (on-shift), FIFO workers in the resource sector may have limited access to healthy food or make poor dietary choices and may engage in limited physical activity. In the resource sector, there have also been reports of risky behaviours, such as excessive alcohol [9] and tobacco consumption [10] at levels that are above those of the general population. A 2020 study [11] of oil and gas workers in the United States reported that the prevalence of excessive alcohol use was 30%. Furthermore, a systematic review of health behaviours in the resource sector, inclusive of the oil and gas industry, found that while on-shift, workers were more likely to drink alcohol excessively, smoke tobacco, and have a poor diet [2]. Frequent travel and being away from home are also related to fatigue and sleep disturbances, with more than 20% of FIFO workers reporting moderate-severe sleep disturbances [12]. FIFO workers generally sleep less and have lower-quality sleep while on-shift compared to when at home [2], which may impact their work performance and safety.

#### 3.2.3. Impact of FIFO Work Arrangements on Mental Health

Mental health concerns have also been documented in FIFO workers, with over 30% of FIFO workers in Australia experiencing psychological distress, compared to just 10% of the general (non-FIFO) population [12]. Poor mental health in FIFO workers may be linked to conflicting home/work demands [2,12], loneliness or isolation [12], and bullying or poor workplace cultures [2]. In contrast, self-reported mental and physical health were better where workers felt they had high levels of social and managerial support [2].

#### 3.2.4. Prevalence of Mental Health Complaints

Even non-FIFO workers in the resource sector have poorer mental health than do workers in other industries and members of the general population. In coal miners, psychological distress (measured via the Kessler Psychological Distress Scale [K10]) was moderate–very high in 39% of workers, which was significantly higher than in the general population, after controlling for age and gender [13]. James et al. reported similar findings, whereby 44% of miners working in remote areas of Australia reported moderate–very high distress (K10 scores) [14]. Furthermore, a broad study of resource workers found that one-third reported moderate–severe depression [15]. Of further concern is that there remains a stigma surrounding mental health in the resource sector, which makes workers less likely to seek support [16]. In mining and construction workers in remote areas of Australia, 40% perceived there was a stigma surrounding mental health and 39% reported that help was unavailable when needed. Importantly, any response from being slightly stressed to being very stressed about the stigma attached to mental health problems was predictive of experiencing high–very high levels of psychological distress (K10; odds ratio = 3.6–23.5) [17]. In combination, the data from the resource sector highlight that mental health is a particular concern for workers, and strategies for addressing this issue should be considered in the provision of healthcare.

#### 3.2.5. Primary Care and Emergency/Trauma Responses

For oil and gas workers, access to primary care is often required before they can return to their usual healthcare service. Evidence relating to primary care needs of workers in the resource sector is limited, and data primarily relate to utilisation of local health services while on-site. Concerns have been raised that an influx of workers to rural and remote communities could strain already-under-resourced local healthcare services [18]. A study conducted in a remote mining community in Queensland, Australia found that up to 30% of health-service and emergency department presentations were from non-residents, and of those, 50% were living in camps [18]. It was also highlighted that over half of the cases from non-residents were non-urgent and could have been treated by a general practitioner [18]. These data echo concerns raised in an Australian Government report [19], namely, that utilisation of health services by non-residents meant that health staff could not meet the resultant demands. It also aligns with a study of community representatives from rural areas hosting oil and gas workers in the United States, who were concerned about reduced access to healthcare and mental health facilities [20]. While it is unclear whether FIFO workers have insufficient access or they actively choose not to utilise healthcare facilities managed by their employer, strategies need to be in place for companies in the resource sector to offer some primary care to employees or to support local health services in being able to meet the increased demand. 

Although the oil and gas industry is highly regulated in terms of health and safety, strategies must also be in place for dealing with emergency or trauma situations in the workplace. Particularly in a remote setting, there may be lengthy delays until professional care can be provided or evacuation to a higher-order health service can be achieved. These low-probability, high-impact incidents subsequently have considerable consequences [21]. In light of this, it is important to explore healthcare delivery in the resource sector as it relates to both emergency/trauma response and primary care. 

Overall, data relating to the health status and needs of oil and gas workers show that poor health behaviours (e.g., diet, exercise, and excessive alcohol consumption) and the consequences associated with these behaviours (e.g., overweight/obesity), as well as sleep disturbances, are prevalent in oil and gas workers, particularly those on FIFO work patterns. It is also clear that the health-related needs of oil and gas workers relate to appropriate access to both primary healthcare and care in the event of an emergency or trauma situation. Nevertheless, some data suggest that workers may inappropriately access local health services, although the reasons for this remain unclear. Therefore, the subsequent sections of this review will explore available evidence regarding healthcare for oil and gas workers and healthcare models from other industries which may be relevant.

### 3.3. Topic 2: Healthcare Delivery

#### 3.3.1. Emergency and Trauma Response

Responding to emergencies and trauma situations is arguably the most critical aspect of healthcare delivery in the oil and gas industry, due to the potential consequences of these events. In an emergency or trauma situation, timely access to first-responder care and emergency patient management may determine patient survival. This can be increasingly difficult in more remote environments, with data showing that the risk of major trauma death is almost double in rural and remote areas compared to metropolitan areas [22]. The first 60 min after a trauma situation, referred to as the golden hour, is often the most critical window in determining an outcome for the patient [23]. For this reason, strategies must be in place for timely patient care, including staff in remote environments being trained in to deliver cardiopulmonary resuscitation (CPR) and use automated external defibrillator (AED), and ensuring an increase in on-site staff able to provide medical care/first aid as remoteness increases [24].

A number of frameworks exist which outline the minimum level of emergency care that must be available. For example, Bonato et al. [25] described the Medical Emergency Response Classification Instrument (MERCI), which was designed in the context of the oil and gas industry. The instrument takes into account the risk (administrative through to industrial), accessibility (remoteness), and population (number of workers) to provide an overall score. The score is then used to guide the determinations of times when support needs to be available (hours per day/at all times), staff required (doctors, nurses or a combination), and materials/equipment/medications which need to be available [25]. The IOGP have a similar framework for medical emergency response and primary care [26]. They propose that healthcare strategies be developed based on risk, healthcare needs, and locally available resources, but with the flexibility to be scaled up and down as needed. The emergency framework is premised on minimising harm and optimising recovery in the event of a medical emergency, and appropriate care is assessed by both quality and the speed at which it can be provided. The framework is separated into four tiers, ranging from first aid (including AED), which should be delivered in around 3–5 min (tier 1), through to referral to a specialist hospital as soon as possible (tier 4). The forementioned frameworks are similar in that they are context-specific and encourage flexibility in how healthcare is delivered during a medical emergency, and as such, it is clear that no single model of healthcare will suit all organisations and contexts. 

#### 3.3.2. Primary Healthcare

Industry bodies such as IOGP have guidelines for the provision of primary care which offer a useful framework for developing healthcare strategies in a manner that is based on risks, healthcare needs, and available resources [26]. Within this framework, primary care typically relates to capacity-building of existing healthcare or establishing a clinical service on or near the work site. Capacity building may involve upskilling local clinicians and/or purchasing equipment to increase diagnostic capacity. On the other hand, a ‘sole proprietor’ model may be employed whereby the organisation builds and operates their own clinic, or they may choose to use a ‘conglomerate model’ by partnering with another company [26]. No model of primary care is considered to be superior, but instead, the ideal model should be developed via a thorough assessment of resources, risk assessments, and insufficiencies of the community’s local primary care services. What is recommended within the framework, however, is that, at times where healthcare needs are lower, other activities may be conducted such as health promotion and occupational health services [26]. This recommendation aligns with a significant volume of published evidence which has demonstrated that preventative healthcare strategies such as interventions targeting behaviour change and employee well-being at the organisation level can have significant benefits to employee wellness and health status [16,27,28].

#### 3.3.3. Preventative Healthcare: Organisation-Led Initiatives

Given the increased likelihood that workers in the resource sector have poor health behaviours, both employees and employers may benefit from the implementation of behaviour-change interventions for workers, such as interventions targeting healthy eating, physical activity, and alcohol consumption [29]. While this does not strictly fall within the scope of primary care, a number of studies have reported that organisation-led interventions and changes to workplace culture have resulted in favourable outcomes for health and well-being in oil and gas and mine workers [16,30,31]. In this regard, organisations such as the National Institute for Occupational Safety and Health (NIOSH), a research agency under the Centre for Disease Control and Prevention (CDC), have developed a strategy for a more holistic and preventative healthcare, called Total Worker Health (TWH), which can be implemented by organisations to support the health of their employees [32]. The development of this strategy stems from the assumption that preventative healthcare can lead to reduced healthcare costs as well as increased productivity [27]. A systematic review of TWH-related interventions found that health behaviours such as diet, physical activity, and tobacco use can be improved, based on data drawn from related industries, including manufacturing and construction [33]. In addition, the IOGP health-management framework [34], which is specific to the oil and gas industry, outlines that risk management strategies should encompass a range of factors including fitness for work, health surveillance, worker well-being, occupational health, fatigue management, management of drug and alcohol misuse, and planning for infectious disease controls. 

In the mining industry, a number of health promotion interventions have been trialled which primarily relate to improved dietary and physical activity behaviours of employees. Khanal et al. [28] described a statewide (NSW, Australia) ‘Get Healthy at Work’ program whereby resources and support can be accessed via an online portal as part of the behaviour-change intervention. The study reflected on initial implementation, but changes in employee health and health behaviours were not assessed. Bezzina et al. [35] also reported on a health and wellness intervention in the mining industry which was an eight-step framework based on a model by the World Health Organisation, though outcomes of the intervention are not yet available. The available evidence suggests that organisations see the value of these programs, and are supportive of implementing more preventative healthcare within the services they offer and, as such, the interventions should be further explored and evaluated.

In contrast, organisation-led interventions focusing on supporting better mental-health-related outcomes have been widely evaluated and endorsed within the resource sector. The interventions which appear to be most successful are those which are driven by the employer, but which create/facilitate peer networks and aim to strengthen non-professional supports for employees [36]. The success of organisation-led mental health interventions likely relates to demonstrated associations between a workplace commitment to mental health and improved employee well-being [37], along with workers in the resource sector preferring to seek mental health support from non-professional (i.e., peers, colleagues, friends, or family) rather than professional supports (i.e., an employee assistance program [EAP] or a general practitioner [GP]) [12,38,39]. A recent systematic review of mental health interventions in the mining industry found consistent associations between employee perceptions that their organisation was committed to good mental health or had organisational support available and improved mental health, well-being, and perceived quality and efficiency of work [16]. Similar findings have been reported whereby miners who perceived that their company was not committed to employee mental health had higher levels of distress [13].

For resource-sector organisations that have delivered and evaluated mental health-related interventions to employees, the results have been positive. Sayers et al. [31] evaluated an intervention for mine workers at two sites called the MATES in Mining (MIM) program, a peer-support program involving general awareness training (all employees), specialist training (to a smaller number of volunteers), and applied suicide intervention skills training (for individuals in a health and safety role at the workplace). Six and eighteen months after the intervention, employees were more confident and likely to seek both professional and non-professional support for mental health, were less likely to agree that they would be treated differently by friends/colleagues if they knew they had a mental illness, and were less likely to think they would be treated poorly in the workplace [31].

While mental-health-related outcomes are important to the individual (employee) it may also be a sustainable investment for the organisation, with poor mental health suggested as being the leading cause of absenteeism, poor work performance, and employee turnover [40]. Ling et al. [41] sampled data from 1456 coal miners which was used to estimate lost productivity in the Australian mining industry due to psychological distress at $154 million per year [41]. In addition, higher psychosocial risk has been seen in mining employees who reported lower job satisfaction and work commitment and higher turnover intention [42].

### 3.4. Topic 3: Innovative Models of Healthcare Delivery

To overcome the challenges in healthcare delivery in the oil and gas industry, there is a need to be both flexible and innovative in delivery of required health services. The most researched innovative model of care incorporates telehealth and telemedicine, of which the value appears well-supported by evidence.

#### 3.4.1. Telehealth and Telemedicine

In 2015, Berg et al. reported on a remote healthcare strategy designed for the delivery of healthcare for offshore oil and gas operations in the Arctic [43]. The innovation of this model related to an integrated system whereby a ‘virtual doctor’s office’ was established on-site, with links to a hospital. In this setting, the virtual doctor’s office included lab testing, X-rays, and ultrasound equipment. The on-site physician then communicated with onshore specialists to support decision-making and patient care. The report highlighted that 22% of medevacs could have been prevented and 72% of medevacs would have led to better and safer patient management if the telehealth system had been used. Similar data reported by Anis et al. [44] found that 39% of patients from offshore facilities could have been treated by telemedicine without being evacuated. Data comparing the frequency of medical evacuations between locations with and without telehealth available reported that the odds of success for medical evacuations were around three times higher (odds ratio = 2.9–3.6) if no telehealth was available [45]. In addition to lower rates of medical evacuations, it has also been reported that oil-rig workers treated via telemedicine were generally satisfied (98%), and employees felt that the provision of telehealth services improved both morale and perceived productivity, as well as making workers feel that the company cared about them being in a remote environment [46].

A number of similar models have successfully utilised other health professionals (non-physicians) to deliver remote healthcare and have also reported favourable outcomes. Eadie et al. [47] explored the feasibility of remotely supported ultrasound completed by paramedics. Paramedics completed ultrasounds in simulated trauma situations and were supported by satellite communications (image transmission and expert guidance). Using this approach, they were able to obtain appropriate scans in 94% of cases [47]. Similar findings were reported by Boniface et al. [48], whereby paramedics without ultrasound experience were able to capture images in a trauma situation in under five minutes, when they had remote guidance. It has also been demonstrated that such systems can be used to support medication delivery. In one example, 14 oil rigs were given thrombolytic drugs. Paramedics on the rigs were trained to deliver the drugs, and the rigs also were supplied with ECG machines which transmitted data via email. After implementation, 47 cases of chest pain were handled via telemedicine over a 3-year period, but only 13% needed evacuation to shore [49]. Latifi et al. [50] report that historically, the use of the telephone for specialist consults meant that patients were often transported to ‘be safe’; however, with recent technological innovations and the consultant being able to see the patient and obtain data (e.g., vitals), decision-making and patient care are improved and transfers may be avoided.

Most data relating to the virtual doctor’s office and similar models have been positive; however, in more extreme remote operations, the possibility for lower-quality telemedicine has been documented [51]. In a systematic review of telemedicine in maritime and extreme weather contexts, it was reported that poor communication networks can impact the quality of telemedicine, and that the distance of medical evacuations, if required, remains a concern [51]. Integrating telemedicine systems into healthcare still presents a number of challenges. These challenges include managing offshore/onshore medical professionals, patient transport, equipment for both collaboration and examinations, rules/regulations, and modifying past ways of working [52]. As such, clear implementation strategies and monitoring processes are essential, and such models should be used to support rather than replace other healthcare strategies [53]. In evaluating the success of telehealth systems in remote operations, no validated indicators have been documented [54]; however, a recent systematic review [54] suggested a number of key performance indicators based on the current evidence. It found that quality telemedicine in a remote setting should include reliable, transmittable, and continuous equipment-based monitoring available within 15 min, high-quality video systems available within 15 min, electronic records accessible via telemedicine, a major hospital available for specialist consulting, appropriately qualified teleconsultant physicians and offshore personnel, clear protocols for managing common cases, and appropriate training of all personnel in using telemedicine [54].

While innovative telehealth/telemedicine models have advantages for the oil and gas industry, research also shows that less-sophisticated, more cost-effective measures can also add value. For example, Mika et al. [55] reported on a range of e-health strategies which had been effective for an international oil and gas company. One approach was a 24 h call centre which connects employees to a doctor via telephone. Also discussed was a health database which supports ready access to confidential health records, along with software for real-time medical stocks (medication/consumables), and a health education portal with resources and e-learning modules for employees. These various strategies provide opportunities for cost-effective solutions which may reduce unnecessary strain on local health services and proactively improve the health behaviours and health status of oil and gas workers.

#### 3.4.2. Public–Private Partnerships

A public–private partnership (PPP) may be proposed as a means of facilitating improved access in smaller rural and remote communities whilst also managing the associated costs. The Institute for Global Health Sciences [56] provides a framework for healthcare PPPs, with key characteristics of PPPs being a long-term contract (a suggested minimum of 5+ years, but typically 15+), shared risk, mutually agreed performance indicators, and public ownership of assets at the end of the contract. Different models may be used depending on whether the model seeks to build/refurbish existing infrastructure (infrastructure-based models) or increase service delivery capacity (discrete clinical service models) or is a combination of infrastructure and capacity building (integrated models). Once the proposed model and the need for the PPP have been established, stages include considering delivery options via feasibility/economic studies, issuing a competitive tender process, and awarding the contract (including negotiating terms and securing funding), after which operations commence. It is also highlighted in the framework that successful PPPs require that the public/private sectors are committed to long-term collaboration, transparency between partners, broad stakeholder engagement, a complete contract which supports changes if required, and public and private sectors having the capacity to deliver [56].

An additional framework specific to the oil and gas industry [57] drew from past examples of healthcare models to provide recommendations for how oil and gas companies can work with local health services. The framework outlines that key stages/considerations included stakeholder consultation/consent, making changes to the health system that could be sustained, and leveraging resources, all of which should occur in distinct phases: data gathering and assessing the situation, situational analysis to determine strengths/weaknesses/feasibility, and forming a strategy which involves joint involvement from the company and local healthcare. Weinberg et al. [58] further suggest that cooperation between oil and gas companies, local government, and other agencies is essential, that industry can help the public sector meet benchmarks, and that stakeholder consultation should occur early.

Despite the availability of various frameworks and recommendations, PPPs for healthcare reveal conflicting outcomes relating to their success or otherwise. Furthermore, much of the evidence is drawn from large-scale projects, such as hospitals, making it difficult to draw clear conclusions relevant to the oil and gas industry. PPPs are proposed as being beneficial, as they combine strengths of both the private (innovation through to entrepreneurship) and public (e.g., responsibility, public accountability) sectors [59]. Infrastructure Partnerships Australia reports that successful PPPs are generally those which can align public/private interests, utilise financial incentives, encourage innovation, and create operational efficiency. Any bids/tenders for PPPs based solely on cost are not advised, but rather innovation and the potential for quality service delivery should be prioritised [60].

On the contrary, a number of groups have raised concerns about PPPs in healthcare, with some going as far as recommending they be abolished entirely [61]. Many argue that given the costs and time associated with contract negotiations, any potential profitability from PPPs is negated. There are also concerns that public and private priorities are incompatible with public healthcare’s need to be accountable to stakeholders and communities rather than private organisations [61]. Further challenges to implementing PPPs relate to government regulations [59], inadequate resources [59,62], poor communication between public/private sectors [59,62], negotiating shared risk [59], inadequate monitoring and evaluation [59], poor management [62], human resource concerns (distrust/power disparity between partners) [62], and IT systems (poor documentation and systems used by the different partners being incompatible) [62].

The administrative burdens associated with PPPs appear to be substantial. For example, Farquhar et al. [63] described a model of physiotherapy care whereby a private physiotherapy organisation serviced hospital patients and aged care residents and the health service provided treatment rooms for them to see patients. The model was deemed successful, as it improved access for patients and satisfied the various stakeholders. However, while it was branded a successful project, they reported a significant administrative burden in contract management/accounting, and a lengthy negotiation was required to finalise the contract. They also reported that managing a private service within a public-healthcare setting created challenges via the uncertainty in the responsibilities for managing the service and contracts. As such, while there is no definitive reason that PPPs should or should not be implemented in rural/remote healthcare services, serious consideration should be given to the question of whether potential advantages outweigh the administrative burden and time taken to finalise contracts.

In addition to the feasibility/practical considerations of models such as PPPs, the evidence also highlights that companies should critically assess how these healthcare models may impact the local communities in the long term. Fluctuations in healthcare access and changes to the local economies where employees (oil, gas, and mining) are coming to work heavily affect more remote communities [64]. While IOGP guidelines [26] recommend that healthcare models in the oil and gas industry be able to be scaled up or down as required, data suggest that future changes to service provision in the local community (i.e., scaling down) likely will have significant impacts. Therefore, any partnerships for the delivery of healthcare should be those which will not negatively impact the local community at the end of the contract period.

## 4. Limitations and Future Directions

Due to the lack of specific evidence regarding ideal models of healthcare in the oil and gas industry, the data for this review have been drawn from a wide range of health and healthcare data in the entire resource sector (mining, oil and gas). As such, findings drawn from the wider resource sector should be interpreted with caution when considering their application to the oil and gas industry. In addition, a narrative approach has been taken in conducting the review. As such, while clearly defined search strategies have been utilised, it is possible that the list of studies included in this review are not exhaustive.

Within the review, a number of areas of future research have also emerged. It is clear that more research is needed to understand health status and healthcare models in the oil and gas industry, encompassing onshore and offshore operations. Preventative healthcare in the form of organization-led interventions were frequently discussed in the literature; however, the effectiveness of interventions relating to lifestyle behaviours (e.g., diet and exercise) have not yet been extensively evaluated. In addition, more research is needed to specifically explore whether public–private partnerships could be a feasible solution for healthcare delivery within the oil and gas industry.

## 5. Conclusions

There is clear evidence that workers in the resource sector are susceptible to poor health behaviours (those relating to diet, exercise, alcohol and tobacco consumption), often have low sleep duration and quality, and have a higher prevalence of mental health concerns than the general population. Further, these behaviours and health concerns appear to be exaggerated in FIFO workers. In addressing poor health behaviours, there is an emerging body of evidence that behaviour change interventions driven by the organisation may be able to improve some of these behaviours. In addition, there is strong evidence that organisation-driven interventions positively impact employee well-being and mental health.

Provision of some primary healthcare is important for FIFO workers, given their time spent away from home. Inappropriate access, such as visiting the emergency department for non-urgent cases which could have been treated by a GP, is otherwise likely. Moreover, the value of organisational-led preventive healthcare and mental health and well-being supports is clear. It is important that organisations have strategies in place for appropriate healthcare or collaboration with local health services which ensure they have sufficient capacity to handle the demand. However, there appears to be no clear answer as to whether partnerships between the public and private sectors (PPPs) may facilitate healthcare access for both resource workers and the local communities. Such arrangements, even on a small scale, are lengthy and complex processes which require significant investments, both to plan and operationalise the partnership, and thus should be critically evaluated/considered.

Emergency and trauma situations are unlikely events, but can have considerable consequences when they do occur, meaning that required healthcare needs to be timely and of a high quality. The most promising innovations for healthcare and emergency responses, in remote areas in particular, are telehealth and telemedicine strategies. So called ‘virtual doctors offices’ have consistently showed positive outcomes for employees and organisations. Such models have led to improvements in timely care and patient satisfaction, as well as fewer patients being medically evacuated from the site. Nevertheless, lower-cost e-health models such as tele/video consults and online resources still appear to be valuable measures for particular aspects of healthcare delivery.

## Figures and Tables

**Table 1 healthcare-11-02888-t001:** Keywords and databases searched for each literature search.

Search Topic	Database	Keywords
Health of employees working in the mining industry and resource sector	PubMed, EMBASE, CINAHL, PsycINFO	#1 Miner, miners, mining industry, resource sector, oil and gas NOT data mining AND #2 physical health, health behavior *, sleep, stress, mental health, fatigue, alcohol, smoking, diet, exercise, distress, sick *, ill *, well-being
Health status of remote and FIFO workers	PubMed, EMBASE, CINAHL, PsycINFO	#1 FIFO, fly-in-fly-out, fly in fly out, DIDO, drive-in drive-out, drive in drive out, long distance commut * AND #2 physical health, health behavior *, sleep, stress, mental health, fatigue, alcohol, smoking, diet, exercise, distress, sick *, ill *, well-being
Healthcare in mining communities	PubMed, EMBASE, CINAHL, PsycINFO, SCOPUS	#1 mining communit *, mining town * AND #2 healthcare, health care, models of care, health service
Healthcare in remote communities	PubMed, EMBASE, CINAHL, PsycINFO, SCOPUS	#1 remote communit *, isolated communit * AND #2 healthcare, health care, models of care, health service
Trauma and emergency response in remote and mining communities	PubMed, EMBASE, CINAHL, PsycINFO, SCOPUS	#1 remote communit *, isolated communit *, mining communit *, mining town * AND #2 Trauma, emergency response, major injur *, advanced life support
Healthcare in the oil and gas industry	PubMed, EMBASE, CINAHL, PsycINFO, Scopus, Google Scholar	#1 oil and gas, energy sector, energy industr * AND #2 healthcare, health care, health service *, models of care, trauma, emergency response, telehealth, telemedicine
Public–private partnerships in healthcare	Google, Google Scholar	#1Public private partnership, health, Australia (AND search) #2 partnership, health, (oil or gas) (AND search)

Note: * Indicates that truncation was used

## Data Availability

Not applicable.

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
