# Peer review of "A Narrative Review of Health Status and Healthcare Delivery in the Oil and Gas Industry: Impacts on Employees, Employers, and Local Communities"

_healthcare, 2023, doi:10.3390/healthcare11212888_

Round 1

Reviewer 1 Report

Comments and Suggestions for Authors Congratulate the authors for the interest in the study carried out. I would make some suggestions to increase the scientific quality of the manuscript.

Methodology:

Specify more relevant details of the methodology: criteria for inclusion and exclusion of articles, people who participate in the review process and phases, specify the information analysis strategy, some instrument has been used to assess the scientific quality of the articles consulted?

 Results: Include a Flow diagram of study selection. Present the characteristics of the studies in terms of country, type of design or study, year of publication, context, and other relevant information. Present in a table the main results regarding the objectives raised in the study (health problems, health-related needs, healthcare delivery and innovative models of care) and according to the contexts analyzed (miner, resource industry, FIFO, communities ).

Topic 1: In general clarify the results to facilitate reading and to clearly identify what the results are in terms of the main health problems and needs of these groups. Furthermore, in this first topic, work factors associated with health problems are identified. Clarify this point. 

Topic 2 and 3: It is not sufficiently clear if the authors are presented in this paper existing care models in these areas or theoretical proposals on the most desirable healthcare delivery or both.

The heading “Primary care and emergency response (line 169) could be part of theme 2 (healthcare delivery) rather than theme 1 (health and needs).

About the topic 3: Could this topic be part of topic 2? In the case of being different, please clarify what the difference is.

Reviewer 2 Report

Comments and Suggestions for Authors

The aim of this paper are twofold; First, to broadly explore health status and needs of workers in the oil and gas industry, healthcare delivery models including primary care and emergency responses, and innovative models of care. Second, to understand how various healthcare models may impact the communities in which the oil and gas workers are based or come to work. Based on the result of this literature review the authors briefly discuss what needs to be considered when designing effective healthcare models for the specific bransch. My main concerns are related to the scientific relevance of this study (as currently motivated in the introduction) and the methodology used. These are commented more in detail under each heading below.

Introduction

The content in the introduction does not help the readers to understand why there is a need to explore the different parts of the aim. For example, the introduction primarily focus on arguing for the relevance of identifying the ideal model of healthcare for oil and gas workers and their employers but does not clearly provide the knowledge gap related to this part of the aim. The reason for why health status and the needs of the workers had to be explored is not argued for at all, i.e. is it a scientific gap that needs to be adressed? If yes, why? I have difficulties understanding the scientific value of the study from the structure of the introduction. Also, very few references are used to build up the case. I would suggest the authors to rewrite the introduction and make sure that it contains information on the problem of interest, what we know from the literature, what we don´t know (the gap) and why we need to know it.

The study consider health status and needs of workers in the oil and gas industry. In the introduction the employers are mentioned together with the concept Healthcare. The concept is not defined but it seems that it includes primary care and emergency care. Occupational health services or similar are not mentioned despite the focus on workers health. This deviates from other studies in the field on worker health. Keywords related to occupational health are lacking in the search strategy. What is the reason for excluding occupational health from the study aim and the searches? How could this affect your result? Another central concept used is healthcare delivery models. What does that include? Please include definitions of central concepts in the paper.

Material and method

The authors have chosen to conduct a narrative review due to the broad scope of interest, the lack of a single specific research question and due to lack of inclusion and exclusion criteria. No specific methodological reference is cited to guide the readers on the methodology used. When checking the scientific literature there seems to lack guidelines on how such a review should be conducted and reported. However, there has been attempts to define best practices, see for example the paper by Rossella Ferrari in Medical writing from 2015. Other authors have also made attempts to define when narrative reviews are appropriate to use. What seems to differ a narrative review from a scoping review, a method suitable when there is an interest in mapping the literature in a given topic and determine possible gaps, is the methodology used to identify the literature and the way the literature is synthesized and discussed. For example, a narrativ review seems to be used when the aim is to describe and discuss a specific topic from a theoretical point of view with and comes with less demands on reporting the methodology used. A scoping review has guidelines describing how to conduct a systematic search to identify the literature. The reporting of the results could be summarized without requirements to discuss them from a theoretical point of view. In the current paper the authors use a comprehensive search stategy reported in the paper. They mention the total number of hits and total number of numbers of papers included after screening. All papers (n=118) are included in Supplementary table 2 with a full reference list at the end of the supplement. Even though this part does not fulfill the methodology criterias of a scoping review the structure reminds more of a scoping review than a narrative review. The results of the search are presented and discussed for different parts of the primary aim. This structure also seem to be more closely related to a scoping review than a narrative review. Especially since the discussion does not add a theoretical contribution to the field and conclusions are sometimes drawn about evidence in certain areas. A reference to how the authors interpret and conduct the narrative review would probably clarify som of the confusion of the methodology used.

Ref: Ferrari, R. Writing narrative style literature reviews. Medical writing, 2015; 24(4); 230-235.

Supplementary table 2 contains more references than the paper itself. I cannot see how the result from S2 was used in the paper, for example which papers were chosen as references and on what grounds. Similarily, I wonder why some of the papers weren´t chosen as references in the result section? If these would have been included, would it have affected the results and conclusions?

Result & discussion

Some of the things brought up in the result section is not related to the aim. For example the association between health status and working conditions and/or work environment factors.

In the result/discussion section, under the subheading Topic 2, preventative healthcare provided by the employer is brought up and discussed. Even though I see the relevance of this section, it has not been brought up in the introduction, in the aim or used as keyterms in the searches. From this perspective, it could be discussed if the studies referred to should have been included in the paper. The current search strategy has probably left out a number of relevant studies, potentially affecting the quality and result of the study.  

It is not defined what is meant by healthcare delivery models but the content in the different sections differ between structural models and planning of health care services, and specific interventions directed towards different health problems. Are interventions included in the concept healthcare delivery models? If not, it should not be included in the result section. Please also see my other comment on potential consequences for the result of the study when relevant keyterms related to occupational health has been left out in the search strategy.

I suggest the authors to avoid using reinforcing words when presenting results, for example ”frequently” (line 144), ”largely” (line 147) and ”well-supported by evidence” (line 315). The latter could be problematic due to the chosen methodology in the paper. More examples are present in the paper.

It seems like the secondary aim has not been adresses in the result/discussion section.

Conclusion

Please check so the text in this section correspond to possible conclusions that could be drawn from the methodology used. Also, check that the conclusion drawn is related to the topic that has been investigated (aim).

Other

The titel does not reflect the content of the study. It is to some extent related to the second aim but the vast majority of the paper is about something else. I recommend the authors to reconsider the titel.

Reviewer 3 Report

Comments and Suggestions for Authors

1. The abstract is very good as it's a concise form of the IMRAD format (Introduction, methods, results and discussions) with a strong conclusive statement from the study.

2. The keywords presented throws a spotlight on all the aspects of the research and would make it easy for researchers to chance on the article when it is eventually published in online databases.

3. The introduction, though concise (as it should be the case of review manuscripts) offers a rich theoretical background to the study, elevating its relevance and presenting its primary and secondary aims.

4. The methods section is satisfactory. The search strategy employed as well as the cited databases are credible. However, the authors must substantiate from the literature, the appropriateness of the narrative review method for this type of review. Moreover, authors must systematically describe how the data garnered from the articles selected were carefully analysed. It would be great for a scholarly manuscript to detail specific data analytical procedure employed with justifications of their choice from the literature.

5. The strongest section of the manuscript is the results and discussion section. It is detailed and discusses intelligently various Healthcare data  aside from the possible health risks faced by stakeholders in the oil and gas industry.

6. The section, limitations and future directions present information on the limitations and not future directions for studies in the area researched. I suggest that the future directions are added to the concluding section. After summarizing the study and its purpose, draw sound conclusions from each of the key findings of the review. Then present recommendations (sound and feasible with implementation actors or agencies) for practice in the oil and gas industries. More importantly, and lacking in this review article, are recommendations for future research. A good review paper must expose various areas for further research. This important ingredient is missing in this review manuscript.

Comments on the Quality of English Language

The manuscript needs to be proofread thoroughly to fix the few spotted errors in syntax.
